# Regional mechanical dyssynchrony and shortened systole are present in people with Takotsubo syndrome

Jan-Christian Reil [1,8] ✉, Vasco Sequeira [2,8], Gert-Hinrich Reil[3], Paul Steendijk [4], Christoph Maack [2], Thomas Fink[5], Elias Rawish[6,7], Ingo Eitel[6,7,8] & Thomas Stiermaier[6,7,8]

## Abstract

**Background:** Takotsubo syndrome is characterized by transient regional systolic dysfunction, left ventricular (LV) dilatation, and edema, often occurring without obstructive coronary artery disease. The mechanisms underlying this stress-induced condition, especially the role of mechanical dyssynchrony in affecting systolic function, remain poorly understood.

**Methods:** In our study, we evaluated global LV function and mechanical dyssynchrony in 24 Takotsubo patients compared to 20 controls by analyzing pressure-volume loops and time-varying elastance. Additionally, we monitored changes in LV segmental volume and internal flow.

**Results:** Here we show a significant reduction in global myocardial contractility and pronounced mechanical dyssynchrony in Takotsubo syndrome, particularly in the mid and apical LV segments, without disturbances in electrical conduction.

**Conclusions:** Our findings reveal substantial mechanical dyssynchrony in Takotsubo patients, characterized by increased internal flow and a shortened systolic ejection time. This indicates a mechanical basis for the inefficient LV function in Takotsubo syndrome, independent of electrical conduction abnormalities.

## Plain Language Summary

People with Takotsubo syndrome have temporarily weakened heart muscle, and this is often triggered by emotional or physical stress. We compared the heart pressure and volume between people with Takotsubo syndrome and healthy individuals. Our findings showed that people with Takotsubo syndrome have weaker heart muscles and irregular contractions, especially in the middle and tip of the heart, despite having normal electrical signals. This means that their hearts are less efficient at pumping blood. This study provides additional details about the impact of Takotsubo syndrome on heart function. Improved understanding of these issues could lead to better diagnosis and treatment for people with this condition in the future.

Heart failure, often a result of combined myocardial irregularities, abnormal loading conditions, and electrical and mechanical nonuniformities, severely impacts the heart's pumping efficiency and increases energy demands. One key mechanical irregularity is left ventricular (LV) dyssynchrony[1]. This dyssynchrony can result from nonuniform electrical and/or mechanical activities across myocardial segments and may influence the evolution of Takotsubo syndrome. Takotsubo, commonly known as 'broken heart syndrome', is characterized by a transient, acute form of heart failure often precipitated by emotional or physical stress[2]. This stress-induced condition manifests with a distinctive pattern of regional ventricular contraction irregularities, leading to characteristic end-systolic ballooning patterns, such as apical, midventricular, basal, or focal hypo- or akinesia[3]. Despite its recognized electrocardiographic changes like T wave inversion, the mechanisms underlying its regional contraction abnormalities and mechanical dyssynchrony remain poorly understood.

The OCTOPUS trial (NCT03726528) previously utilized the conductance catheter methodology to assess pressure-volume (PV) characteristics of Takotsubo patients' LV[4]. These findings highlighted Takotsubo's reduced contractility, with a shortened systolic period (dTes = reduced time to end-systolic elastance) and a rightward shift in the PV diagram, indicative of Frank-Starling mechanism activation to sustain stroke volume[4]. These changes were accompanied by decreased stroke work and mechanical

[1]Klinik für allgemeine und interventionelle Kardiolgie, Herz-und Diabetes-Zentrum Nordrhein-Westphalen, Georgstrasse 11, 32545 Bad Oeynhausen, Germany. [2]DZHI, Department of Translational Science Universitätsklinikum, Würzburg, Germany. [3]Universitätsklinik für Innere Medizin – Kardiologie, Klinikum Oldenburg, Rahel Strauss Strasse 10, Oldenburg, Germany. [4]Department of Cardiology, Leiden University Medical Center, Leiden, the Netherlands. [5]Clinic for Electrophysiology, Herz- und Diabeteszentrum NRW, Ruhr-Universität Bochum, Bad Oeynhausen, Germany. [6]Medical Clinic II, University Heart Center Lübeck, Lübeck, Germany. [7]German Center for Cardiovascular Research (DZHK), Partner site Hamburg - Kiel -Lübeck, Lübeck, Germany. [8]These authors contributed equally: Jan-Christian Reil, Vasco Sequeira, Ingo Eitel, Thomas Stiermaier. ✉e-mail: janchristian.reil@gmail.com

efficiency[4]. Despite our understanding of global PV characteristics, the nuanced distribution patterns of regional hypokinesia in Takotsubo syndrome are yet to be fully explored. The present study aims to advance knowledge in this area through a detailed analysis of the OCTOPUS trial data, focusing on regional mechanical dyssynchrony across five myocardial segments from the LV apex to base, and comparing them to controls. We show that Takotsubo patients exhibit significant mechanical dyssynchrony, particularly in the mid and apical LV segments, accompanied by reduced contractility and a shortened systolic ejection time. This mechanical inefficiency occurs without disturbances in electrical conduction, revealing a primarily mechanical basis for impaired LV function in Takotsubo syndrome.

## Methods
### Study design, participants, and analysis
The OCTOPUS study (Unique identifier: NCT03726528), as detailed in Stiermaier et al[4]., included patients with Takotsubo syndrome (TTS) and clinical controls, documenting their comorbidities and medication histories. The diagnosis of TTS conformed to both the European consensus criteria[5] and the more recent, comprehensive InterTAK criteria[6], which encompass (1) transient regional LV wall motion abnormalities, usually extending beyond a single epicardial vascular distribution; (2) absence of obstructive coronary artery disease or other pathologic conditions that could account for the observed pattern of temporary LV dysfunction; (3) new and reversible electrocardiographic abnormalities; (4) elevated serum levels of natriuretic peptides with minor cardiac troponin elevation; and (5) recovery of LV systolic function on follow-up imaging. Patients with atrial fibrillation were excluded from the study due to the challenges associated with analyzing PV relationships in the presence of irregular cardiac cycles, which can introduce considerable variability in the measured parameters and hinder the determination of consistent average values. The study initially included 25 patients, who underwent comprehensive hemodynamic assessment with PV loop recordings. One patient was excluded post hoc because of persistent wall motion abnormalities and the detection of myocardial scar tissue on cardiac magnetic resonance, resulting in a final cohort of 24 patients with confirmed TTS. Reflecting the recognized sex prevalence in Takotsubo syndrome, our cohort consisted predominantly of postmenopausal women, with 23 out of 24 patients being female. The average age of our cohort was $70.6 \pm 12.6$ years, which aligns with the age distribution commonly seen in this condition. For ethical reasons, instead of healthy individuals, 20 patients with clinically suspected coronary artery disease were selected as controls. They underwent an invasive assessment of PV relationships using the same standardized protocol that was applied to the TTS group. These patients required coronary angiography (e.g., stable angina pectoris, dyspnea) and qualified as active participants in the control group, provided they had no major cardiovascular disorders (e.g., stenosing coronary artery disease, valvular heart disease, heart failure, pulmonary hypertension, heart rhythm disease). The control group comprised an equal number of male and female participants, with an average age of $57.1 \pm 7.2$ years, significantly younger than the TTS patient cohort ($p = 0.0009$). These controls were part of another study conducted at our institution titled 'Effects of acutely elevated afterload on left ventricular contractility and relaxation in heart failure with preserved and reduced ejection fraction' (ANREP-EF; Unique identifier: NCT02751853), which was discontinued. The consent provided by the participants in the ANREP-EF study also covered their inclusion in the OCTOPUS trial. All patients with TTS and approximately 50% of the control group were on β-blockers. The data collection for the OCTOPUS trial involving Takotsubo patients took place at the University Heart Center Lübeck, Germany, from October 2018 to May 2022. Similarly, the control group data from the ANREP-EF study were collected at the same center between January 2017 and May 2022. The study protocols for both the OCTOPUS trial and the ANREP-EF study were reviewed and approved by the local ethics committee at the University of Lübeck,

Germany. Prior informed consent was obtained from all patients whose data was included in the OCTOPUS trial and the ANREP-EF study.

### Pressure–volume loop recordings and calibration
LV function in Takotsubo syndrome patients (TTS, $n = 24$) was compared to that of controls ($n = 20$) using continuous pressure and volume signals obtained via 7F combined pressure-conductance catheters (CD Leycom, Zoetermeer, The Netherlands) connected to a Cardiac Function Lab (CFL-512, CD Leycom). Analysis focused on cases with a minimum of four or five ventricular segmental volume signals, leading to inclusion of 22 TTS (out of 24, 14 with apical and 8 with midventricular Takotsubo) and 14 control participants (out of 20). In the remaining cases, due to the smaller size of the heart, it was not possible to position the required minimum of four conductance catheter segments within the LV. This limitation prevented adequate analysis of dyssynchrony in these instances. The baseline hemodynamic parameters are presented in Table 1. Steady-state PV loops were calibrated to match angiographically-determined end-diastolic (EDV), stroke volume (SV) and end-systolic volumes (ESV) before the procedure[7]. Given the emergent and acute settings under which our Takotsubo patients were examined, often during nighttime conditions in the cardiac catheterization laboratory, angiographic calibration was chosen due to its rapid and reliable measurement capabilities. This method allowed simultaneous calibration with the conductance measurements, aligning with the need for immediate assessment under these conditions[8,9]. Using a single-beat estimation, the end-systolic PV relationship (ESPVR) was determined and characterized by its slope ($E_{es}$) and volume-axis intercept ($V_0$)[7]. We adopted the time-varying elastance approach to examine the temporal evolution of LV contractility. Instantaneous elastance $E(t)$ was calculated at each PV loop point by connecting $V_0$ to that specific PV point: $E(t) = P(t)/[V(t) - V_0]$. For each heartbeat, elastance–time curves were plotted and end-systole was defined as the moment of the peak of this curve ($E_{es}$). We then defined the duration of systole as the time interval between end-diastole (defined by the R-wave of the QRS complex) and end-systole (time to end-systolic elastance = dTes). Using the first derivative of the elastance-time curves, we measured the maximum increase ($dE/dt_{max}$) and decrease ($dE/dt_{min}$) rates. Additionally, the time between $dP/dt_{max}$ and $dP/dt_{min}$ was calculated to measure the systolic period. Recognizing the significant influence of heart rate on systolic duration[10], we applied the Fridericia formula derived for the electrocardiographic QT interval correction ($QT_c = QT/(RR^{1/3})$) to obtain a

## Table 1 | Basic hemodynamic parameters in Takotsubo (TTS) patients and controls

| | TTS ($n = 22$) | Control ($n = 14$) | $p$-value |
|---|---|---|---|
| Heart rate (bpm) | $84 \pm 16$ | $72 \pm 11$ | 0.03 |
| QRS complex (ECG) (ms) | $95 \pm 8$ | $93 \pm 10$ | 0.50 |
| Time $dp/dt_{max}$ to $dp/dt_{min}$ (HR corrected) (ms) | $327 \pm 26$ | $371 \pm 34$ | <0.0001 |
| dTes (ms) | $288 \pm 34$ | $337 \pm 23$ | <0.0001 |
| dTes$_c$ (ms) | $317 \pm 28$ | $359 \pm 19$ | <0.0001 |
| Ejection fraction (%) | $0.48 \pm 0.10$ | $0.62 \pm 0.07$ | 0.0001 |
| $E_{es}$ (mmHg/ml) | $1.7 \pm 1.01$ | $3.0 \pm 1.5$ | 0.004 |
| dp/dt$_{max}$ (mmHg/s) | $1467 \pm 347$ | $1768 \pm 416$ | 0.03 |
| Ea (mmHgml) | $2.12 \pm 0.98$ | $1.96 \pm 0.48$ | 0.56 |
| End-diastolic volume (ml) | $144 \pm 54$ | $107 \pm 21$ | 0.04 |
| End-systolic volume (ml) | $73 \pm 26$ | $40 \pm 8$ | 0.002 |
| Stroke volume (ml) | $71 \pm 35$ | $66 \pm 17$ | 0.65 |
| IF (ml) | $37 \pm 22$ | $22 \pm 5$ | 0.013 |
| IFF (%) | $42 \pm 13$ | $29 \pm 9$ | 0.002 |

Data are presented as mean ± standard error of the mean (SEM)

*TTS* Takotsubo syndrome, *HR* heart rate, *dTes* duration of systole, *dTes$_c$* corrected systolic ejection time, *Ees* end-systolic elastance, *dp/dt$_{max}$* maximal rate of pressure rise, *Ea* effective arterial elastance, *IF* internal flow, *IFF* internal flow fraction.

standardized (i.e. at 60 bpm) systolic duration[11]. This provided a corrected dTes termed $dTes_c$. Indicators of contractility included $E_{es}$ and $dP/dt_{max}$, while effective arterial elastance (Ea) was used as an indicator of afterload.

## Assessment of mechanical dyssynchrony

Our present study conducts a retrospective analysis of pressure-volume (PV) loop data, emphasizing mechanical dyssynchrony across five left ventricular segments. This innovative approach expands upon the initial goals of the OCTOPUS study (NCT03726528), which sought to elucidate the pathophysiology of Takotsubo Syndrome through an in-depth hemodynamic assessment, focusing on PV loops and the dynamics of systolic and diastolic functions. The conductance catheter allows for real-time measurement of five segmental volume slices ($V_{seg}$) perpendicular to the LV long axis. The first slice corresponds to the apex, while the fifth associates with the base of the heart (see Fig. 1, inset Fig. 2). At each time point, a segmental signal was defined as dyssynchronous if its change

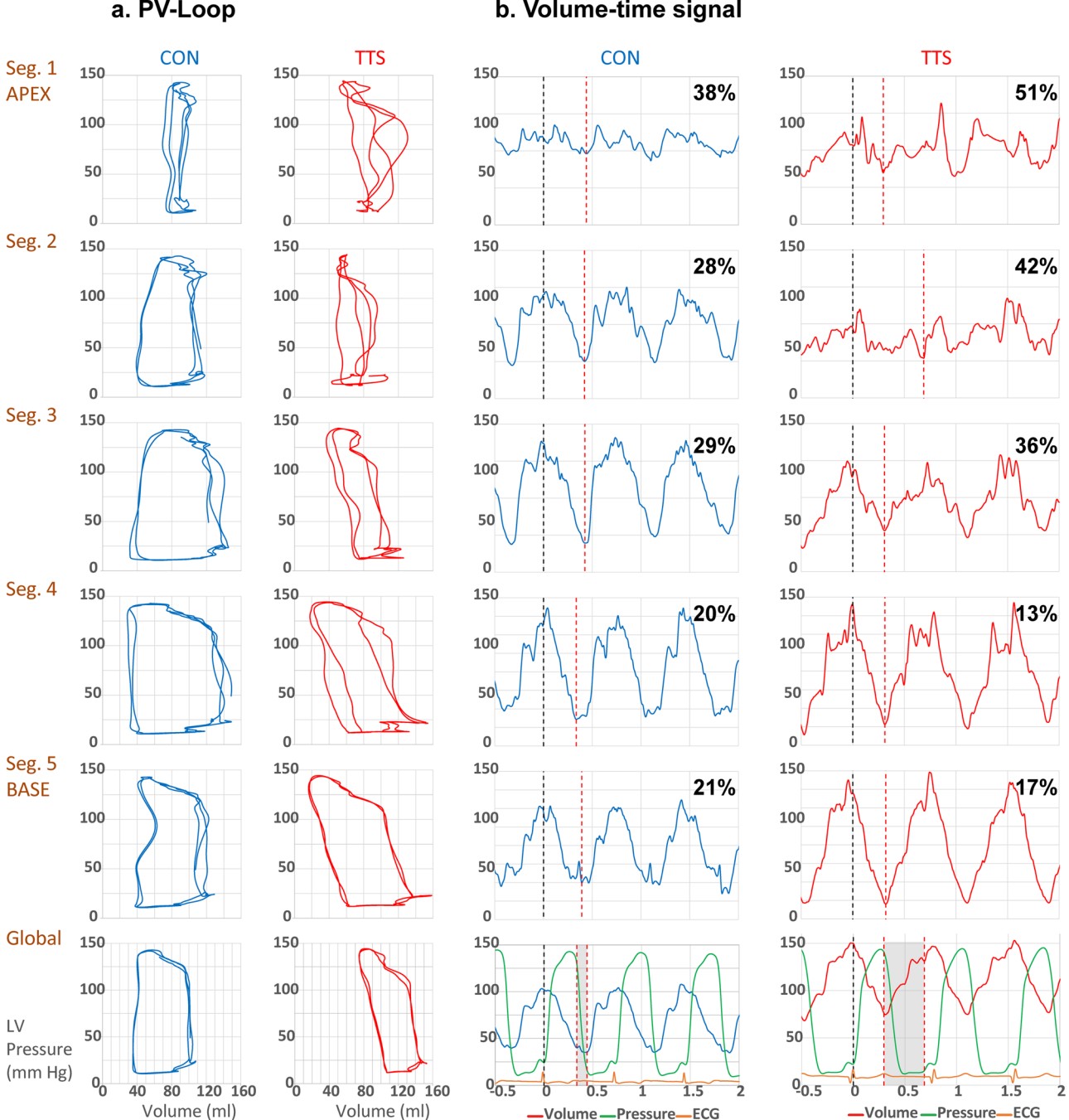

**Fig. 1 | Pressure–volume and volume–time characteristics in control and Takotsubo patients. a** Typical segmental 1 (apex) to 5 (base) pressure-volume (PV) loops and the global PV loop of a control (blue) and a Takotsubo (TTS) patient (red). The *Y*-axis represents pressure in mmHg, and the *X*-axis volume in ml. **b** Typical segmental 1 (apex) to 5 (base) volume-time diagrams and global volume-time diagram for a control (blue) and a Takotsubo (TTS) patient (red). The *Y*-axis measures volume in ml, and the *X*-axis time in ms. In each segment, the time from maximal ($V_{max}$ = start point) to minimal volume ($V_{min}$) is indicated by black and red dotted lines, respectively. The global volume–time diagram also includes the pressure-time signal in green. The gray shaded area within the global volume-time signal illustrates the spread of time up to $V_{min}$ across all 5 segments, which is markedly more pronounced in the Takotsubo patient compared to the control, indicative of dyssynchrony. The percentage values in the individual volume-time diagrams reflect the dyssynchrony of the respective segment when it is out of phase with the global segment. Notably, these values are increased in the apical region in TTS.

**Fig. 2 | Assessment of dyssynchrony metrics in Takotsubo Syndrome and control patients.**
Patients with Takotsubo syndrome (TTS, $n = 22$) are depicted in red and controls (Co, $n = 14$) are shown in blue. **a** illustrates the Internal Flow Fraction (IFF), and (**b**) shows the Internal Flow (IF). **c** Depicts segmental dyssynchrony %, while (**d**) demonstrates the time to minimal segmental volume ($V_{min}$). **e** Standard deviation (SD) of $V_{min}$ across all five segments for both TTS patients and controls. Exact *p*-values are presented, and statistical significance is assumed when $p < 0.05$, compared to controls. The inset *p*-values represent one-way ANOVA analysis for intersegmental variations. Data are presented as mean ± standard error of the mean (SEM). The inset diagram provides a schematic of the conductance catheter's positioning within the left ventricle (LV), delineating the segments from apex (segment 1) to base (segment 5).

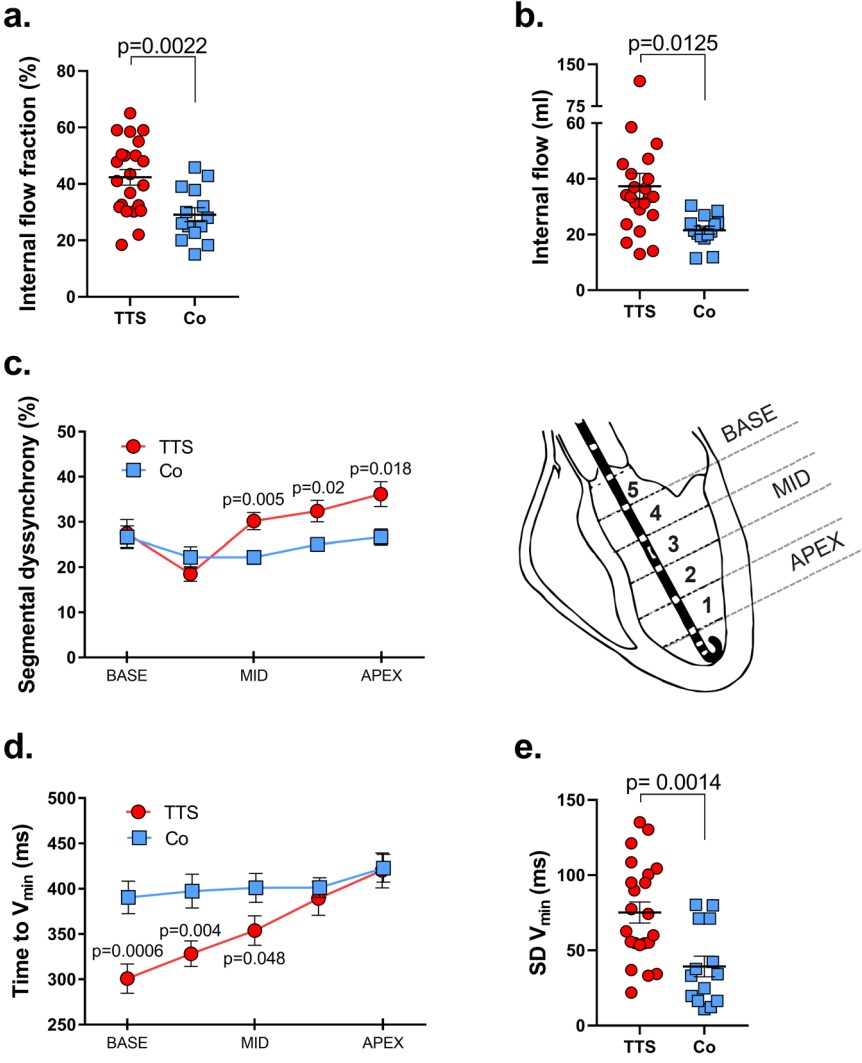

(i.e., $dV_{seg}/dt$) was opposite to the simultaneous change in the total LV volume ($dV_{LV}/dt$) of the heart. Subsequently, segmental dyssynchrony is quantified by calculating the percentage of time within the cardiac cycle during which a segment is dyssynchronous. The average of the segmental dyssynchronies was taken as a measure of global LV dyssynchrony. An additional marker of dyssynchrony involves determining the time taken for each segment to reach its respective minimal volume ($T_{seg, Vmin}$), which serves as an indicator of the individual segment's end-systole, and calculating the standard deviation of the five segments. To calculate the individual segmental systolic time after aortic valve closure (=post-ejection time), the difference between $T_{Vmin}$ and dTes was taken ($T_{Vmin}$−dTes). During this time difference, intersegmental volume changes occurred while the aortic valve remained closed, and the total volume within the entire LV was unchanged (global isovolumetric relaxation). Nonuniform contraction and filling leads to inefficient shifting of blood volume within the LV. This internal flow (IF) is quantified by calculating the sum of the absolute volume changes of all segments and then subtracting the absolute total LV volume change: IF(t) = [Σ |dVseg(t)/dt−dV_{LV}(t)/dt|]/2. Note that $dV_{LV}(t)/dt$ represents the effective flow into or out of the LV[1]. Thus, IF measures the segment-to-segment blood volume shifts, which do not result in effective filling or ejection. Division by two takes into account that any ineffective segmental volume change is counterbalanced by an equal but opposite volume change in the remaining segments[1]. The IF fraction (IFF) is calculated by integrating IF(t) over the full cardiac cycle, normalized against absolute effective flow[1].

## Statistics and *R*eproducibility
Continuous variables were analyzed using unpaired Student's t-test or Mann-Whitney-test, depending on distribution normality, with results presented as mean ± standard error of the mean (SEM). Intersegmental changes within groups were assessed by one-way ANOVA. A *p*-value < 0.05 indicated statistical significance. All analyses were performed using GraphPad Prism 8.1.2.

## Results
### Segmental and global LV function in Takotsubo syndrome: a detailed analysis through PV loops
Figure 1 illustrates typical PV loops and volume-time traces for a patient with Takotsubo syndrome alongside a control subject. These are presented across five heart segments, from the apex (segment 1) to the base (segment 5). Notably, the segmental PV loops clearly show the inefficient ventricular pump function of the Takotsubo patient especially in the apical segments, with abnormal isovolumic contraction and relaxation phases (Fig. 1). Additionally, there is a noticeable variability in the time taken to reach the minimum volume ($V_{min}$) in Takotsubo patients

**Table 2 | Segmental dyssynchrony (%)**

| | Seg 5 (basis) | Seg 4 | Seg3 | Seg 2 | Seg 1 apex |
|---|---|---|---|---|---|
| **Dys (%)** | | | | | |
| TTS | 27.5 ± 13.3 | 18.5 ± 7.6 | 30.2 ± 9.0 | 32.4 ± 11.1 | 36.2 ± 12.9 |
| Control | 26.7 ± 9.2 | 22.2 ± 8.8 | 22.2 ± 5.6* | 25.0 ± −5.4* | 26.7 ± 6.3* |
| **Time to $_{vmin}$ (ms)** | | | | | |
| TTS | 301 ± 70 | 328 ± 66 | 354 ± 76 | 389 ± 87 | 420 ± 91 |
| Control | 392 ± 67* | 400 ± 70* | 403 ± 61* | 404 ± 41 | 425 ± 57 |
| **ΔT$_{Vmin}$ - dTes (ms)** | | | | | |
| TTS | 21 ± 54 | 49 ± 57 | 74 ± 68 | 108 ± 78 | 144 ± 70 |
| Control | 38 ± 40 | 35 ± 44 | 44 ± 47 | 53 ± 39* | 81 ± 47* |

Segmental time to minimal volume (Vmin) and segmental difference of time to $V_{min}$ ($T_{Vmin}$) and duration of systole (dTes).

Data are presented as mean ± standard error of the mean (SEM).

*TTS* Takotsubo syndrome, *dTes* duration of systole, $V_{min}$ minimal volume, $TV_{min}$ time to minimal volume.

*$p < 0.05$ vs control. Exact *p*-values for comparisons can be found in Figs. 2c, d, and 5.

compared to controls (Fig. 1). Global LV PV loops in 22 Takotsubo patients and in 14 controls confirm a reduction of myocardial contractility in Takotsubo syndrome. This is evidenced by reduced end-systolic elastance ($E_{es}$) and $dP/dt_{max}$, and diminished ejection fraction (EF), which align with higher end-systolic and end-diastolic volumes (Table 1). Notably, this decline in contractility occurs even under comparable afterload conditions, as reflected by unaltered effective arterial elastance (Ea) (Table 1).

### Dyssynchrony and internal flow alterations in Takotsubo syndrome: evidence of inefficient cardiac mechanics

Previous research on heart failure patients has revealed that nonuniform contraction and filling lead to an ineffective shift of blood volume within the LV, especially during the phases when both the aorta and mitral valves are closed, such as isovolumic contraction and relaxation periods[1]. This phenomenon is termed "internal flow" (IF) or, when expressed as a percentage of the net global flow, the internal flow fraction (IFF)[1]. Our study's detailed analysis of dyssynchrony metrics showed increased IFF and IF in Takotsubo patients compared to controls (Table 1). This indicates pronounced intersegmental blood volume shifts which do not contribute to ejection. This phenomenon results in inefficient blood movement, akin to a "dead space" shift or a form of wasted ejection work (Fig. 2a, b). In the controls, IF primarily occurs during the isovolumic contraction and relaxation phases, which is consistent with normal physiology.

Mechanical dyssynchrony in Takotsubo patients is further evidenced by increased segmental dyssynchrony, particularly in the apical and mid-ventricular segments (segments 1 to 3) (Fig. 2c and Table 2). These segments frequently move out of phase with the global heart rhythm, often corresponding with apical or midventricular ballooning patterns (Fig. 2c). Sub-analyses of Takotsubo phenotypes (apical [n = 14] vs. midventricular [n = 8]) reveal that apical Takotsubo significantly influenced our overall findings (Supplementary Fig.). These patients exhibited notable dyssynchrony in segments 1–3 (apex to mid) compared to corresponding segments in controls and significant intersegmental asynchrony between segments 1–3 and 4–5 (mid to base). In contrast, midventricular Takotsubo patients also showed significant dyssynchrony in segment 3, though the intersegmental differences were less pronounced, likely due to the smaller sample size (Supplementary Fig.).

Figures 1, 2d, e, and Table 2 illustrate a notable delay in reaching minimum volume across segments in Takotsubo patients compared to the synchronous timing in controls. This results in a significantly higher standard deviation of $T_{Vmin}$ in the Takotsubo group, indicative of larger dyssynchrony (Fig. 2e). The similarity in QRS complex durations between Takotsubo patients and controls rules out electrical conduction abnormalities (Table 1).

### Altered systolic timing and contractility in Takotsubo syndrome: insights from systolic period and ejection dynamics

A consistent finding in all Takotsubo patients was the shortening of systolic period, as indicated by the reduced time between $dP/dt_{max}$ and $dP/dt_{min}$ (Fig. 3c), and the time to $E_{es}$ (Fig. 3d). End-systolic elastance (slope $E_{es}$) and systolic ejection time (time to $E_{es}$), both indicators of LV contractility, decrease in patients with Takotsubo syndrome (Table 1). In Takotsubo patients, both dTes (Fig. 4a) and dTes$_c$ (Fig. 4b) were significantly shortened. $dE/dt_{max}$ did not differ between groups (Fig. 4c), while $dE/dt_{min}$ was significantly smaller (more negative) in Takotsubo compared to controls (Fig. 4d). The post-systolic ejection time ($T_{Vmin}−$dTes) was considerably prolonged in the mid and apical segments of Takotsubo patients, and intersegmental systolic post-ejection times were significantly lengthened between base and apical segments in both groups (Fig. 5 and Table 2).

### Discussion

Our previous research characterized the global LV function in Takotsubo syndrome, and identified reduced cardiac contractility, inefficient myocardial energetics, and prolonged active relaxation, alongside unaltered diastolic passive stiffness[4]. The current investigation extends these observations by examining regional hypokinesia across five myocardial segments, from the LV apex to the base. Through an analysis of segmental volume signals, PV loops, and elastance-time evaluations, we uncovered substantial LV mechanical dyssynchrony in Takotsubo patients, especially in the mid and apical segments. Notably, this dyssynchrony occurs without associated electrical conduction disturbances, differentiating Takotsubo syndrome from other forms of acute heart failure that typically exhibit dyssynchrony alongside electrical alterations such as left bundle branch block[1].

The presence of mechanical dyssynchrony in the apical and midventricular segments is underscored by various metrics: an increased percentage of time that a segment remains dyssynchronous relative to the global signal, an increased standard deviation of the time to reach minimum volume (Fig. 2e), and prolonged post-systolic ejection time (Fig. 5 and Table 2). These findings, aligning with Takotsubo syndrome's typical ballooning patterns, emphasize the impact of mechanical nonuniformities. Additionally, elevated IF and IFF underscore a shift in internal blood volume, indicative of nonuniform contraction and filling (Fig. 2a, b), thereby supporting a global reduction in LV function. Interestingly, the control group exhibited higher-than-expected IFF values (mean value 29%), which more closely resembled those of patients with coronary artery disease (CAD, 20%)[1], suggesting a less healthy cohort than anticipated despite the exclusion of stenosing CAD, valvular heart disease, and the preservation of LV systolic function. Nevertheless, Takotsubo patients showed significantly higher IFF values (42%) and other dyssynchrony parameters, reinforcing the mechanical nature of their condition's dyssynchrony, independent of

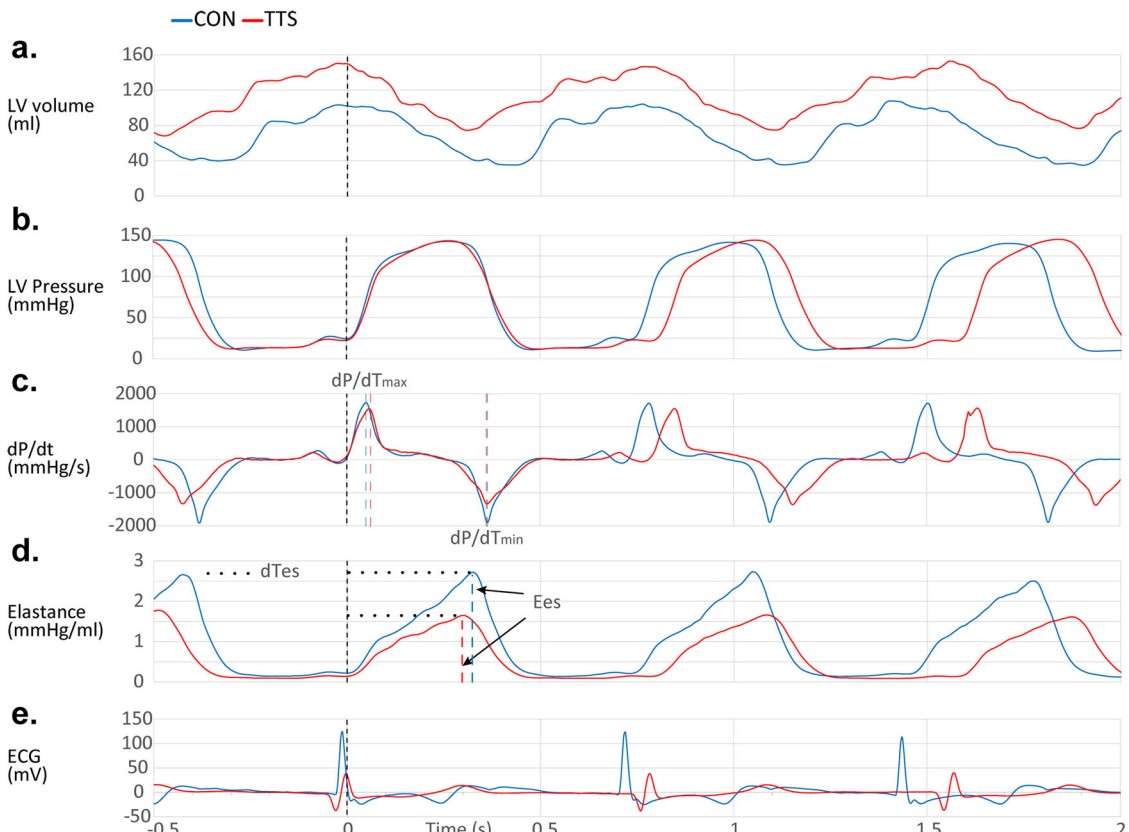

**Fig. 3 | Comparative traces between a control subject (blue) and a Takotsubo syndrome patient (red). a** Representative volume-time traces, illustrating the heart's volume changes over time. **b** Pressure–time traces, indicating the intraventricular pressure throughout the cardiac cycle. **c** d$p$/d$t$ traces, which reflect the rate of pressure change within the ventricle, a key indicator of cardiac contractility. **d** Elastance–time traces, demonstrating the heart's ability to contract and relax, while (**e**) presents the electrocardiogram traces, showing the electrical activity of the heart. Although the second heartbeat for both subjects starts simultaneously, as evidenced by the dashed line marking $t = 0$, the time d$p$/d$t_{max}$ to d$p$/d$t_{min}$ (depicted with dashed vertical red lines) is shorter in the Takotsubo patient. In addition, decreased end-systolic elastance ($E_{es}$, depicted with dashed vertical red lines) and a shortened time to reach end-systolic elastance (dTes, depicted with horizontal dotted black lines) are observed despite a lower heart rate. The shorter systolic period in the Takotsubo patient may be obscured by the slower heart rate; however, when corrected for heart rate (dTes$_c$), the systolic ejection times are 356 ms for the control and 324 ms for the Takotsubo patient, indicating significant functional disparity. TTS, Takotsubo Syndrome; dp/dt, Rate of pressure change; d$p$/d$t_{max}$, Maximum rate of pressure rise during systole; d$p$/d$t_{min}$, Minimum rate of pressure decline during diastole; Ees, End-systolic elastance; dTes, Duration of systole; dTes$_c$, Corrected systolic ejection time.

electrical abnormalities (Fig. 2a). All examined patients had normal QRS complexes, with no evidence of left bundle branch block (LBBB), commonly associated with dyssynchrony (Table 1). When comparing Takotsubo patients to those with heart failure with reduced ejection fraction (HFrEF), who typically show compromised LV systolic function (mean EF of 26%) and LBBB (LBBB, QRS width on average 186 ms), Takotsubo patients demonstrate similar, though less severe, patterns of dyssynchrony[1]. This is noteworthy as dyssynchrony in patients with LBBB is often treated with biventricular pacemakers[12], a strategy that would likely not be effective in Takotsubo syndrome due to its distinct pathophysiological basis. Despite the anticipated lower stroke volumes in Takotsubo syndrome due to decreased contractility our findings showed no significant difference between the stroke volumes of Takotsubo patients and controls (Table 1). This can be explained by the compensatory utilization of the Frank-Starling mechanism in Takotsubo patients, leading to increased LV end-diastolic pressure and volume, which helps maintain stroke volume despite impaired contractility[4].

In assessing Takotsubo syndrome's contractile dynamics through elastance-time curves and their first derivatives, we gained additional information into the potential underpinnings of observed dyssynchrony absent electrical abnormalities. These derivatives reveal the dynamics of the maximal rate of elastance increase (d$E$/d$t_{max}$) and relaxation (d$E$/d$t_{min}$) rate.

In scenarios where increased contractility is accompanied by a shortened time to $E_{es}$ (dTes) and a simultaneous increase of both d$E$/d$t_{max}$ and d$E$/d$t_{min}$, this suggests influences such as adrenergic stimulation, as seen with agents like dobutamine[13]. Interventions like $Ca^{2+}$ sensitizers or myosin activators augment contractility by prolonging dTes without altering the slopes of d$E$/d$t_{max}$ and d$E$/d$t_{min}$[13]. Additionally, we discovered that an increased afterload triggers the Anrep effect, which enhances contractility through the activation of 'dormant' myosin motors and prolongs myosin-actin interactions[14,15]. These mechanisms amplify myosin's stroke work capacity, augment LV contractility, and extend dTes[14,15]. This process is influenced by the phosphorylation state of sarcomeric proteins, especially myosin light chain 2 (MLC2) and cardiac myosin-binding protein-C (cMyBP-C)[14]. In our analysis of Takotsubo patients, we observed a reduced time to $E_{es}$ (Table 1) and decreases in both d$E$/d$t_{max}$ (Fig. 4c) and d$E$/d$t_{min}$ (Fig. 4d), as well as a decrease in contractility expressed by $E_{es}$ with the same afterload (see Ea, Table 1) compared to controls, possibly indicating an inability to utilize the Anrep effect effectively[14,15]. We posit that the underlying cellular mechanisms influencing the reduced contractility and mechanical dyssynchrony observed in Takotsubo progression may involve disruptions in the phosphorylation patterns of these sarcomeric proteins. This is likely related to the process of ventricular torsion, also known as the 'wringing' motion of the heart, which is crucial for effective blood pumping

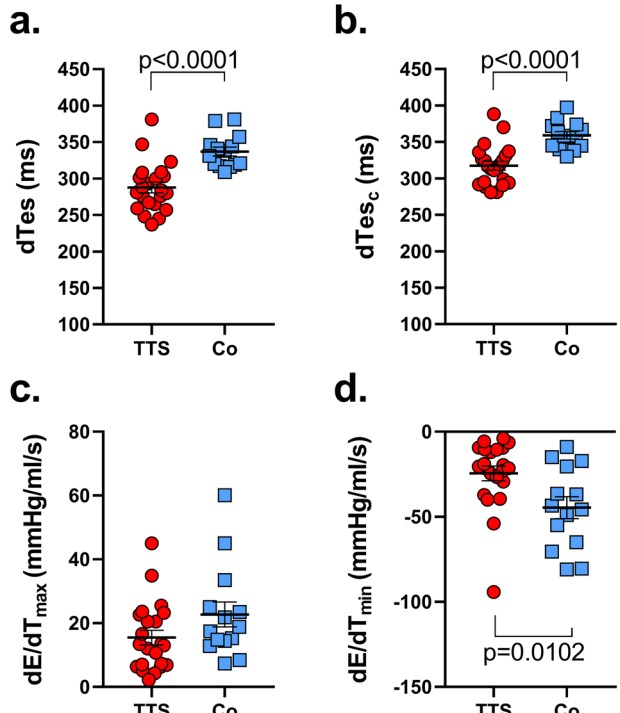

**Fig. 4 | Systolic function parameters in Takotsubo Syndrome and control patients.** Patients with Takotsubo syndrome (TTS, $n = 22$) are depicted in red and controls (Co, $n = 14$) are shown in blue. **a** Duration of systole (dTes), reflecting the duration of blood ejection from the left ventricle. **b** Systolic ejection time corrected for heart rate (dTes$_c$), offering a heart rate-independent comparison. **c** Maximal increase of the elastance–time curves ($dE/dt_{max}$), a measure of the maximum rate at which the ventricle's stiffness increases during systole. **d** Minimal decrease of the elastance-time curves ($dE/dt_{min}$), indicating the rate at which the ventricle's stiffness decreases during relaxation. Exact p-values are presented, and statistical significance is assumed when $p < 0.05$, compared to controls, suggesting notable variations in the systolic function between the two groups. Data are presented as mean ± standard error of the mean (SEM).

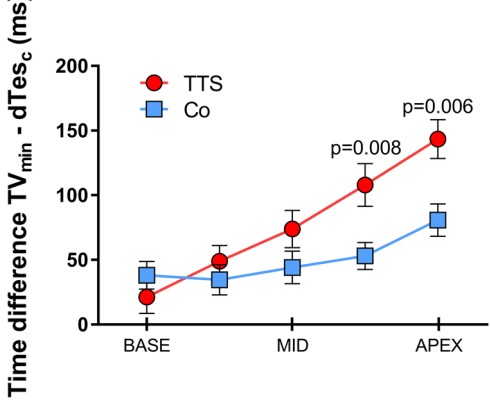

**Fig. 5 | Segmental post-systolic contraction durations in Takotsubo Syndrome and controls.** This graph illustrates the duration of post-systolic volume decrease across different segments of the heart from base to apex. It compares the time difference between the minimal segmental volume (TV$_{min}$) and the duration of systole (dTes) in patients with Takotsubo syndrome (TTS, $n = 22$; red) and control subjects (Co, $n = 14$; blue). Exact p-values are presented, and statistical significance is assumed when $p < 0.05$, compared to controls, obtained through one-way ANOVA for intersegmental comparisons within each group. Data are presented as mean ± standard error of the mean (SEM).

and depends on the helical pattern of cardiac muscle fibers wrapping around the heart[16]. Synchronized electrical signals initiate this process; the base of the heart starts to contract while the apex begins to twist. This twisting motion is complemented by the thickening and posterior movement of the interventricular septum after LV depolarization. As the ventricles contract, the apex rotates against the stationary base, enhancing the squeezing action necessary to propel blood forward[16]. During repolarization, marked by the T-wave on an electrocardiogram, a shiver-like untwisting motion starts at the apex and extends to the base, allowing the ventricles to relax and the septum to revert to its original position and thickness[17].

In Takotsubo syndrome, LV dysfunction occurs independently of electrical conduction abnormalities, suggestion intrinsic myofilament alterations that compromise ventricular torsion. Central to this torsion is the spatial gradient of sarcomeric protein phosphorylation, especially of MLC2[18]. This MLC2 phosphorylation gradient is most pronounced at the heart's apex to facilitate stronger contraction and decreases towards the base, ensuring coordinated myofibril activation and optimizing the wringing motion during systole[18]. The molecular mechanisms that propagate this gradient are not fully described but likely involve differential activation of myosin light chain kinase (MLCK) or Ca$^{2+}$/calmodulin-dependent protein kinase II (CaMKII) enzymes, which phosphorylate MLC2 across different heart regions. Takotsubo syndrome may involve changes in the expression, distribution, and activity of these kinases, as well as in the phosphatases that dephosphorylate MLC2, affecting the phosphorylation pathway from the apex to the base of the heart. Most importantly, the changes in Takotsubo syndrome may be related to alterations in Ca$^{2+}$ dynamics and myocardial energetics, which are known to cause irregular sarcomere activation and dyssynchronous contractions[19]. Emotional stress, often a precursor to Takotsubo syndrome, can trigger spikes in adrenergic activity, leading to Ca$^{2+}$ overload and subsequent impacts on Ca$^{2+}$ handling proteins through cAMP-mediated activation of protein kinase A[20]. Paradoxically, this Ca$^{2+}$ overload coincides with reduced force generation in Takotsubo-induced pluripotent stem cell-derived cardiomyocytes[20]. Similarly, a Takotsubo rat model has shown Ca$^{2+}$ handling defects and energetic deficits, indicating an energy supply and demand mismatch during stress, despite intact mitochondrial function[21]. These observations align with the limited myocardial energetics seen in Takotsubo patients[4]. Since CaMKII phosphorylation of MLC2 is influenced by Ca$^{2+}$ levels and oxidative stress[14], both linked to Takotsubo syndrome[21], it is plausible that these factors negatively affect the MLC2 phosphorylation gradient and ventricular torsion, ultimately contributing to LV dysfunction in Takotsubo patients.

Several limitations must be considered in our study. While the conductance catheter methodology offers high-resolution, real-time data, it is invasive. Its applicability in routine clinical practice may therefore be limited. Although the sample size in our study, might seem limited, it is crucial to recognize the rarity of Takotsubo syndrome. Securing a cohort of 22 patients with such a rare condition is a considerable achievement and strengthens the value of our observations. Nevertheless, future studies with larger cohorts, if feasible, are necessary to validate and build upon our results.

In conclusion, our comprehensive analysis of Takotsubo syndrome underscores the profound mechanical dyssynchrony that characterizes this condition, setting it apart from other forms of heart failure. Unique to Takotsubo syndrome is its pronounced mechanical dyssynchrony, occurring in the absence of electrical conduction delays. Notable features of this syndrome include reduced contractility, shortened systolic period, marked inefficient internal LV flow and diminished mechanical efficiency. These findings contribute to a better understanding of the underlying mechanisms in Takotsubo patients and may guide future therapeutic approaches.

## Data availability

The source data for each figure in this study can be found in the supplementary data files: Supplementary Data 1 contains the data underlying Fig. 2, Supplementary Data 2 contains the data for Fig. 4, Supplementary Data 3 contains the data for Fig. 5, and Supplementary Data 4 contains the

data for the Supplementary Fig.. Data collection for this study pertains to the OCTOPUS trial, registered with the identifier NCT03726528, and the ANREP-EF trial, with the identifier NCT02751853. Additional datasets generated and analyzed during the course of this study, including data from both trials, are available upon reasonable request. These datasets, which are not publicly accessible due to ethical considerations, can be shared in compliance with participant consent and institutional approval. Specifically, deidentified individual participant data, including text, tables, figures, and appendices, will be made available for sharing. Related documentation, such as the study protocol and statistical analysis plan, will also be accessible. Data will be made available immediately following publication with no end date. Researchers interested in accessing the data must submit a methodologically sound proposal, which will undergo review and approval by an independent committee ("committee intermediary"). Data will be shared to facilitate research aligned with the approved proposal, and access will be provided through our university's institutional data repository.

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

## Acknowledgements
We acknowledge support by the Open Access Publication Funds of the Ruhr-Universität Bochum.

## Author contributions
Jan-Christian Reil: Responsible for organizing the study protocol, acquiring and analyzing the data, interpretation, writing, and revising the manuscript. Vasco Sequeira: Analyzed the data, interpretation, writing, and revising the manuscript. Gert-Hinrich Reil: Interpretation of data, writing, and revising the manuscript. Paul Steendijk: Analyzed the data, interpretation of data, writing, and revising the manuscript. Christoph Maack: Interpretation of data, writing, and revising the manuscript. Thomas Fink: Interpretation of data, writing, and revising the manuscript. Elias Rawish: Responsible for organizing the study protocol, acquiring and analyzing the data, interpretation, writing, and revising the manuscript. Ingo Eitel: Responsible for organizing the study protocol, acquiring and analyzing the data, interpretation, writing, and revising the manuscript. Thomas Stiermaier: Responsible for organizing the study protocol, acquiring and analyzing the data, interpretation, writing, and revising the manuscript.

## Funding

## Competing interests
The authors declare the following competing interests: Christoph Maack is an advisory board member for Bristol Myers Squibb, Boehringer Ingelheim, AstraZeneca, Servier, Amgen, Novo Nordisk, Bayer, Novartis, Edwards, and Berlin Chemie. No other conflicts of interest were reported by the remaining authors.
