## [Peer Review File · Communications Medicine]

Reviewers' comments:

Reviewer #1 (Remarks to the Author):

Manuscript summary

Many thanks to the editors for asking me to review this fascinating manuscript, and congratulations to the authors for the in-depth analysis and continued use of valuable, invasively obtained haemodynamic data.

This is a post-hoc analysis of pressure-volume catheter data taken from the OCTOPUS study (published by Stiermaier et al. in JACC 2023) and compared to pressure-volume catheter data obtained from the discontinued ANREP-EF study (NCT02751853). OCTOPUS focussed on physiological changes during Takotsubo cardiomyopathy, and participants' data were, as in the current manuscript, compared against 'active comparators'. This study's hypotheses centre around intersegmental differences in parameters. The occurrence of these findings, without evidence of electrical conduction disturbance, is taken as potential evidence of myocardial cellular impairment, itself caused by disruption in sarcomeric proteins' phosphorylation patterns.

Overall impression of the work

This analysis makes up for limited numbers of participants – understandable given the rarity of Takotsubo cardiomyopathy – with high levels of detail. It extends previous work in the field by the authors and leverages invasively obtained data that may otherwise have remained redundant – for those reasons the authors should be congratulated.

These results are a valuable addition to the armamentarium of invasively obtained haemodynamic theory around Takotsubo cardiomyopathy. Combined with further studies to prospectively examine the correlations explored here, test cellular hypotheses, or include more clinically applicable testing modalities (e.g. echocardiograph/MRI), these data could lead to changes in clinical practice – though that objective remains distant.

Specific comments

I will address methods prior to results, as that would be the usual pattern of manuscript organisation (I appreciate this change may have been an editorial decision).

1. Line 263. Lyon et al. (Eur J H Fail) is quoted as the criteria for Takotsubo cardiomyopathy diagnosis. However Ghadri et al. (EHJ 2018) suggest the InterTAK criteria more recently, whilst also making note of several alternatives: the Mayo Clinic Diagnostic criteria; Japanese guidelines; Gothenburg criteria; Johns Hopkins criteria; Takotsubo Italian Network proposals; and the Madias recommendations. Perhaps a reason for adopting the Heart Failure Association's Taskforce 2016 recommendations is advisable given that context?

2. Line 272. OCTOPUS excluded patients in atrial fibrillation, a common condition and a limitation of the generalisability of the current study to all patients with Takotsubo cardiomyopathy: explicit mention of that exclusion criterion either here or in the limitations would add clarity.

3. Line 288. More detail on 'control' participants' backgrounds would be helpful. Which of the ANREP-EF's three study groups were they part of? What was the indication for their cardiac catheterisation? How did the variability of the PV loop data in the ANREP-EF HFPEF group 1 – mentioned explicitly as the cause for terminating ANREP-EF on clinicaltrials.gov – affect this analysis?

4. Line 309. In the current manuscript the calibration method for the pressure-volume catheter mentioned is angiographic, and Steendijk et al. 2008 is referenced. Yet in Steendijk et al., the calibration of the pressure-volume catheter was performed by transthoracic echocardiography, and was amongst patients with significant hypertrophic cardiomyopathy undergoing alcohol septal ablation. Are there references for your calibration methods that use angiography, and do so among healthier, more comparable samples?

5. Line 322. $dT_{esc} = dT_{es} / RR^{1/3}$ leverages the Fridericia formula for QT duration correction – do you have any evidence of previous applications of electrocardiographic formulae in haemodynamic contexts? Do you need to correct haemodynamic intervals in the same way we do for electrocardiographic ones?

6. Lines 333-355. Given the reference to Steendijk et al. 2004, was mechanical dispersion as a measure of dyssynchrony considered as a potential variable of interest?

7. Line 105 and figure 2A, B, E (and elsewhere). Graphical representations of data that show the complete set (e.g. 'violin' charts) are preferable to 'dynamite' bar charts.

8. Line 123 and figure 3C. It's not clear that there are significant differences in the 'period of systole' (i.e. the duration from $(dP/dt)_{max}$ to $(dP/dt)_{min}$) from Figure 3C, as the ECG illustrates a differing heart rate – perhaps this could be illustrated with lines demarcating the durations for each cycle?

9. Line 123 and figure 3D. Same as point 8, except for Ees (i.e. peak E(t)).

10. Lines 126-129. Same as point 7. The fact that figure 4C and 4D are statistically not-significant and significant respectively illustrates the difficulty with 'dynamite' charts.

11. Line 129. Typographic error – figure 4C should read 4D.

12. Line 132 and table 2 and figure 5. Was there any analysis by Takotsubo phenotype (i.e. apical vs. mid vs. basal variants)? It would seem individual segmental analysis as per table 2 and figure 5 would be confounded by this categorisation, though the transition to greater dyssynchrony towards the apex suggests the cohort is predominantly apical in phenotype.

Reviewer #2 (Remarks to the Author):

I read with interest this original article and congratulate with the authors for providing new pathophysiological insights into this syndrome.

Major comments:

- The authors find through an analysis of segmental volume signals, PV loops, and elastance-time evaluations, significant LV mechanical dyssynchrony especially in the mid and apical segments. However they included in the analysis not only the 13 patients with apical but also 9 pts with midventricular Takotsubo. How the authors explain the base-apex gradient in patients with mid-ventricular pattern? I would report the two subgroups separately or include only those with apical ballooning pattern.
- How was the Stroke Volume calculated? Using volumes or LVOT vti? I would expect lower stroke volumes in TTS patients. How the authors explain the no difference between the two groups?

Reviewer #1:

This is a post-hoc analysis of pressure-volume catheter data taken from the OCTOPUS study (published by Stiermaier et al. in JACC 2023) and compared to pressure-volume catheter data obtained from the discontinued ANREP-EF study (NCT02751853). OCTOPUS focussed on physiological changes during Takotsubo cardiomyopathy, and participants' data were, as in the current manuscript, compared against 'active comparators'. This study's hypotheses centre around intersegmental differences in parameters. The occurrence of these findings, without evidence of electrical conduction disturbance, is taken as potential evidence of myocardial cellular impairment, itself caused by disruption in sarcomeric proteins' phosphorylation patterns.

This analysis makes up for limited numbers of participants – understandable given the rarity of Takotsubo cardiomyopathy – with high levels of detail. It extends previous work in the field by the authors and leverages invasively obtained data that may otherwise have remained redundant – for those reasons the authors should be congratulated. These results are a valuable addition to the armamentarium of invasively obtained haemodynamic theory around Takotsubo cardiomyopathy. Combined with further studies to prospectively examine the correlations explored here, test cellular hypotheses, or include more clinically applicable testing modalities (e.g. echocardiograph/MRI), these data could lead to changes in clinical practice – though that objective remains distant.

Response:

Thank you for your insightful comments and commendations on our manuscript. We have carefully considered your feedback and made revisions to enhance the clarity and impact of our work. We hope that these revisions not only address your concerns but also strengthen the overall quality and contribution of our research. Newly added text is underlined in light blue, and sections removed for clarity are struck through.

Reviewer comment 1:

Line 263. Lyon et al. (Eur J H Fail) is quoted as the criteria for Takotsubo cardiomyopathy diagnosis. However Ghadri et al. (EHJ 2018) suggest the InterTAK criteria more recently, whilst also making note of several alternatives: the Mayo Clinic Diagnostic criteria; Japanese guidelines; Gothenburg criteria; Johns Hopkins criteria; Takotsubo Italian Network proposals; and the Madias recommendations. Perhaps a reason for adopting the Heart Failure Association's Taskforce 2016 recommendations is advisable given that context?

Response:

We appreciate your comment and agree that several diagnostic criteria are available. When designing the OCTOPUS trial, we decided to include patients based on the criteria published in the European Journal of Heart Failure in 2016. Patient enrollment began in 2018, prior to the publication of the InterTAK criteria by Ghadri et al. (EHJ 2018). However, there are only minor differences between these diagnostic criteria. We reviewed our population using the InterTAK criteria and found that all included patients also met these criteria. This aspect has been addressed in the revised manuscript. This change is reflected on page 9, lines 280 to 284 of the methods section of the revised manuscript:

“The OCTOPUS study (Unique identifier: NCT03726528), as detailed in Stiermaier et al.⁴, included patients with Takotsubo syndrome (TTS) and clinical controls, documenting their comorbidities and medication histories. The diagnosis of TTS conformed to both the European consensus criteria¹⁵ and the more recent, comprehensive InterTAK criteria¹⁶, which encompass encompassing:”

Reviewer comment 2:

Line 272. OCTOPUS excluded patients in atrial fibrillation, a common condition and a limitation of the generalisability of the current study to all patients with Takotsubo cardiomyopathy: explicit mention of that exclusion criterion either here or in the limitations would add clarity.

Response:

Patients with atrial fibrillation were not included in the study because the irregular cardiac cycles associated with atrial fibrillation make pressure-volume analysis considerably more challenging. The variability in heartbeats introduces greater variance in the parameters being investigated, complicating the determination of comparable average values over several cycles. In response to your comment, we have revised the manuscript to explicitly mention this exclusion criterion. The following sentence has been added to page 10, lines 290 to 294 of the revised manuscript:

“Patients with atrial fibrillation were excluded from the study due to the challenges associated with analyzing PV relationships in the presence of irregular cardiac cycles, which can introduce significant variability in the measured parameters and hinder the determination of consistent average values.”

Reviewer comment 3:

Line 288. More detail on ‘control’ participants’ backgrounds would be helpful. Which of the ANREP-EF’s three study groups were they part of? What was the indication for their cardiac catheterisation? How did the variability of the PV loop data in the ANREP-EF HFPEF group 1 – mentioned explicitly as the cause for terminating ANREP-EF on clinicaltrials.gov – affect this analysis?

Response:

The ANREP-EF study was initiated to investigate the Anrep effect—the afterload-dependent increase in contractility—in clinical settings. The study aimed to demonstrate this effect in control participants without severe disease and to document its attenuation in patients with HFpEF (Heart Failure with preserved Ejection Fraction) or HFrEF (Heart Failure with reduced Ejection Fraction). Unfortunately, we observed that only one-third of the control participants exhibited the Anrep effect following a handgrip exercise. The remaining control participants either demonstrated an increased adrenergic response or showed increased preload utilization through the Frank-Starling mechanism. Due to the heterogeneous responses within the control group, the ANREP-EF study was discontinued, and no further patients were recruited for the HFpEF or HFrEF groups.

The control group for the OCTOPUS study, as detailed on page 10 of the manuscript, was drawn from this original ANREP-EF control group. These participants had indications for cardiac catheterization such as stable angina pectoris and dyspnea, and they were required to have no significant cardiovascular disorders, including stenosing coronary artery disease, valvular heart disease, heart failure, pulmonary hypertension, or heart rhythm disease. They served as control subjects under resting conditions, which is crucial for ensuring comparability with the Takotsubo patients, who were also assessed under resting conditions.

Under resting conditions, the control participants exhibited consistent and stable hemodynamic parameters that align with those of control groups in other conductance catheter studies. The variability noted in the control group during the handgrip exercise did not affect the resting-state analysis because the controls demonstrated uniform hemodynamic responses at rest.

Therefore, despite the heterogeneous responses observed during handgrip exercise, the ANREP-EF control group remains suitable for comparison with the Takotsubo patients under resting conditions.

Reviewer comment 4:

Line 309. In the current manuscript the calibration method for the pressure-volume catheter mentioned is angiographic, and Steendijk et al. 2008 is referenced. Yet in Steendijk et al., the calibration of the pressure-volume catheter was performed by transthoracic echocardiography, and was amongst patients with significant hypertrophic cardiomyopathy undergoing alcohol septal ablation. Are there references for your calibration methods that use angiography, and do so among healthier, more comparable samples?

Response:

The decision to utilize angiographic calibration was driven by the specific circumstances under which we conducted our measurements. Our patients with Takotsubo syndrome were examined in the cardiac catheter laboratory under emergency conditions, often during night hours. These circumstances, which differ significantly from the well-planned alcohol septal ablation procedures for patients with hypertrophic obstructive cardiomyopathy, necessitated a rapid and reliable calibration method that could be performed simultaneously with the conductance measurements.

There is literature supporting the use of angiography for measuring ventricular volumes. For instance, Dodge and Sheehan (JACC, 1983; doi:10.1016/s0735-1097(83)80012-6) and Dodge and Baxley (Am J Cardiol, 1969; doi:10.1016/0002-9149(69)90006-x) discuss the efficacy of angiographic calibration in obtaining accurate left ventricular volume measurements. Given the emergency and acute settings of our patient examinations, angiographic calibration was deemed the most appropriate solution, guided by the need for immediate assessment and the technical feasibility required during acute patient care.

To account for these changes, we have added new text to the methods section on page 11, lines 333 to 338,

“Given the emergent and acute settings under which our Takotsubo patients were examined, often during nighttime conditions in the cardiac catheterization laboratory, angiographic calibration was chosen due to its rapid and reliable measurement capabilities. This method allowed simultaneous calibration with the conductance measurements, aligning with the need for immediate assessment under these conditions.”^{17,18}

Reviewer comment 5:

Line 322. $dT_{esc} = dT_{es} / RR^{1/3}$ leverages the Fridericia formula for QT duration correction – do you have any evidence of previous applications of electrocardiographic formulae in haemodynamic contexts? Do you need to correct haemodynamic intervals in the same way we do for electrocardiographic ones?

Response:

Recent research by Morbach et al. (European Heart Journal, 2023; <https://doi.org/10.1093/ehjimp/qyad020>) provides a comprehensive study of 4836 participants from the general population to identify heart rate-corrected systolic ejection time (dT_{esc}) by echocardiography. This study demonstrates the utility of the Fridericia-like approach for correcting the duration of systole based on heart rate, thus providing a robust framework for its

application in haemodynamic studies. Morbach et al. derived reference values for dTesc and explored its prognostic utility in patients with acute heart failure, supporting the relevance and accuracy of this correction method.

Given this evidence, we have included a reference to Morbach et al.'s work in our revised manuscript (now reference 21) to substantiate our approach to heart rate correction of dTesc.

Reviewer comment 6:

Lines 333-355. Given the reference to Steendijk et al. 2004, was mechanical dispersion as a measure of dyssynchrony considered as a potential variable of interest?

Response:

We did consider the mechanical dispersion measure proposed in Steendijk et al., but we found it to be complex and not yet validated for our specific patient cohort. Therefore, we preferred to calculate the times to minimal volume (Vmin) for each segment and then specify the corresponding standard deviation (SD), as shown in Figure 2D and 2E. This approach is simpler and closely aligns with the methods widely used in echocardiography to quantify dyssynchrony.

Reviewer comments 7 and 10:

#7. Line 105 and figure 2A, B, E (and elsewhere). Graphical representations of data that show the complete set (e.g. 'violin' charts) are preferable to 'dynamite' bar charts.

#10. Lines 126-129. Same as point 7. The fact that figure 4C and 4D are statistically not-significant and significant respectively illustrates the difficulty with 'dynamite' charts.

Response:

Given the interconnected nature of comments #7 and #10, we have combined our responses to address these concerns. We acknowledge the limitations of 'dynamite' bar charts, which often do not adequately represent the variability and distribution of data points. In response, we have revised Figures 2 and 4 to replace the 'dynamite' bar charts with scatter plots. Scatter plots provide a clearer visualization of individual data points as shown below:

Revised Figure 2.

And regarding Figure 4, please note that Figure 4D was(is) statistically significant.

Revised Figure 4.

Reviewer comments 8 and 9:

#8. Line 123 and figure 3C. It's not clear that there are significant differences in the 'period of systole' (i.e. the duration from (dP/dt)_{max} to (dP/dt)_{min}) from Figure 3C, as the ECG illustrates a differing heart rate – perhaps this could be illustrated with lines demarcating the durations for each cycle?

#9. Line 123 and figure 3D. Same as point 8, except for Ees (i.e. peak E(t)).

Response:

We understand that your comments #8 and #9 are interlinked, and we have therefore combined our responses to them. In line with your suggestion, we have added lines illustrating the durations of each cycle, demarcating dP/dt_{max} and dP/dt_{min}, as well as Ees. These changes are shown below and have been included in Figure 3.

Revised Figure 3.

These adjustments can be found in the revised manuscript (Figure Legend 3) on pages 14 to 15:

“Figure 3. Comparative traces between a control subject (blue) and a Takotsubo syndrome patient (red). (A) Representative volume-time traces, illustrating the heart's volume changes over time. (B) Pressure-time traces, indicating the intraventricular pressure throughout the cardiac cycle. (C) dp/dt traces, which reflect the rate of pressure change within the ventricle, a key indicator of cardiac contractility. (D) Elastance-time traces, demonstrating the heart's ability to contract and relax, while (E) presents the electrocardiogram traces, showing the electrical activity of the heart. Although the second heartbeat for both subjects starts simultaneously, as evidenced by the dashed line marking $t=0$, the Takotsubo patient exhibits slower rates of pressure change (time dp/dt_{max} to dp/dt_{min} , depicted with dashed vertical red lines), decreased end-systolic elastance (E_{es} , depicted with dashed vertical red lines), and a shortened time to reach end-systolic elastance (dT_{es} , depicted with horizontal dotted black lines) despite a lower heart rate. The shorter systolic period in the Takotsubo patient may be obscured by the slower heart rate; however, when corrected for heart rate (dT_{esc}), the systolic ejection times are 356 ms for the control and 324 ms for the Takotsubo patient, indicating significant functional disparity.”

Reviewer comment 11:

Line 129. Typographic error – figure 4C should read 4D.

Response:

Thank you for pointing out this error. We have corrected the reference to the figure on line 141 (page 5) from 4C to 4D in the revised manuscript.

Reviewer comment 12:

Line 132 and table 2 and figure 5. Was there any analysis by Takotsubo phenotype (i.e. apical vs. mid vs. basal variants)? It would seem individual segmental analysis as per table

2 and figure 5 would be confounded by this categorisation, though the transition to greater dyssynchrony towards the apex suggests the cohort is predominantly apical in phenotype.

Response:

We thank the reviewer for the valuable suggestion and have performed additional sub-analyses based on Takotsubo phenotypes (now included as Supplementary Figure and shown below for your reference). Specifically, we calculated segmental dyssynchrony in 14 patients with apical Takotsubo and 8 patients with midventricular Takotsubo. We would also like to clarify that our previous version contained a typographical error regarding the patient numbers; we had originally noted 13 patients with apical Takotsubo and 9 patients with midventricular Takotsubo. Upon careful re-evaluation, we corrected this to 14 and 8, respectively. This correction does not change our findings or their interpretation, as reflected in the main manuscript (line 326). The results indicate that the largest subgroup, patients with apical Takotsubo, significantly influenced the overall findings of our study. These patients exhibited notable dyssynchrony in segments 1-3 (apex to mid; n=14) when compared to the corresponding segments in controls, as well as significant intersegmental dyssynchrony between segments 1-3 and segments 4-5 (mid to base).

Supplementary Figure A. Segmental Dyssynchrony in Takotsubo Syndrome. Segmental dyssynchrony comparison between apical Takotsubo syndrome (TTS) patients and controls (Co). The graph illustrates the percentage of segmental dyssynchrony in apical TTS patients (n=14) and controls across different heart segments: base (5 to 4), mid (3), and apex (2 to 1). Apical TTS patients exhibit significantly higher segmental dyssynchrony in the mid and apical segments compared to controls, indicated by an asterisk (*) for significant differences ($p<0.05$). The overall segmental dyssynchrony in apical TTS patients shows a notable increase from base to apex, suggesting a gradient of mechanical inefficiency progressing towards the apex. Statistical analysis shows a significant difference in dyssynchrony between TTS and controls (TTS $p<0.001$, Co $p=0.29$).

The smaller group of patients with midventricular Takotsubo also demonstrated significant dyssynchrony in segment 3 (mid) compared to controls. Although the highest intersegmental differences in dyssynchrony were observed in the middle and apical segments, they did not reach statistical significance, likely due to the smaller sample size (n=8).

B. Midventricular TTS vs Controls

Supplementary Figure B. Segmental Dyssynchrony in Takotsubo Syndrome. Segmental dyssynchrony comparison between midventricular TTS patients and controls. The graph displays the percentage of segmental dyssynchrony in midventricular TTS patients (n=8) and controls across the base, mid, and apex segments. Midventricular TTS patients show significantly higher dyssynchrony in the mid segment compared to controls. Although there is an observable trend of increased dyssynchrony towards the apex, the difference is less pronounced compared to apical TTS patients, potentially due to the smaller sample size (TTS p=0.085, Co p=0.29).

Furthermore, the distribution of dyssynchrony in both Takotsubo groups appears to be particularly pronounced in segments 1-3 (apex to mid). This might be related to the fact that the boundaries of the affected areas ("Takotsubo area") do not precisely align with the catheter segments, leading to partial fusion of healthy and affected areas, especially in the midventricular Takotsubo group (segments 1-2; apex). This boundary issue could result in mixed signal areas that affect the dyssynchrony measurements.

To reflect these adjustments, we have added the following text to the results section on pages 4 to 5, lines 116 to 123,

"Sub-analyses of Takotsubo phenotypes (apical [n=14] vs. midventricular [n=8]) reveal that apical Takotsubo significantly influenced our overall findings (Supplementary Figure). These patients exhibited notable dyssynchrony in segments 1-3 (apex to mid) compared to corresponding segments in controls and significant intersegmental asynchrony between segments 1-3 and 4-5 (mid to base). In contrast, midventricular Takotsubo patients also showed significant dyssynchrony in segment 3, though the intersegmental differences were less pronounced, likely due to the smaller sample size (Supplementary Figure)."

Reviewer #2:

I read with interest this original article and congratulate with the authors for providing new pathophysiological insights into this syndrome.

Response:

Thank you for your thoughtful evaluation and the positive remarks on our manuscript. We have carefully considered all the feedback received and have made corresponding revisions to enhance the clarity and impact of our work. In the revised manuscript, newly added text is highlighted in light blue, and text that has been removed for clarity is shown with a strikethrough.

Reviewer comment 1:

The authors find through an analysis of segmental volume signals, PV loops, and elastance-time evaluations, significant LV mechanical dyssynchrony especially in the mid and apical segments. However they included in the analysis not only the 13 patients with apical but also 9 pts with midventricular Takotsubo. How the authors explain the base-apex gradient in patients with mid-ventricular pattern? I would report the two subgroups separately or include only those with apical ballooning pattern.

Response:

Thank you for the insightful comment. In response, we have conducted additional sub-analyses based on Takotsubo phenotypes, which are now included as Supplementary Figure and shown below for your reference. We calculated segmental dyssynchrony in 14 patients with apical Takotsubo and 8 patients with midventricular Takotsubo. We would also like to clarify that our previous version contained a typographical error regarding the patient numbers; we had originally noted 13 patients with apical Takotsubo and 9 patients with midventricular Takotsubo. Upon careful re-evaluation, we corrected this to 14 and 8, respectively. This correction does not change our findings or their interpretation, as reflected in the main manuscript (line 326). The results indicate that the largest subgroup, patients with apical Takotsubo, significantly influenced the overall findings of our study. These patients exhibited notable dyssynchrony in segments 1-3 (apex to mid) compared to the corresponding segments in controls, as well as significant intersegmental dyssynchrony between segments 1-3 and segments 4-5 (mid to base).

Supplementary Figure A. Segmental Dyssynchrony in Takotsubo Syndrome. (A) Segmental dyssynchrony comparison between apical Takotsubo syndrome (TTS) patients and controls (Co). The graph illustrates the percentage of segmental dyssynchrony in apical TTS patients (n=14) and controls across different heart segments: base (5 to 4), mid (3), and apex (2 to 1). Apical TTS patients exhibit

significantly higher segmental dyssynchrony in the mid and apical segments compared to controls, indicated by an asterisk (*) for significant differences ($p < 0.05$). The overall segmental dyssynchrony in apical TTS patients shows a notable increase from base to apex, suggesting a gradient of mechanical inefficiency progressing towards the apex. Statistical analysis shows a significant difference in dyssynchrony between TTS and controls (TTS $p < 0.001$, Co $p = 0.29$).

In contrast, midventricular Takotsubo patients also showed significant dyssynchrony in segment 3 (mid), although the intersegmental differences were less pronounced, likely due to the smaller sample size ($n = 8$).

Supplementary Figure B. Segmental Dyssynchrony in Takotsubo Syndrome. (B) Segmental dyssynchrony comparison between midventricular TTS patients and controls. The graph displays the percentage of segmental dyssynchrony in midventricular TTS patients ($n = 8$) and controls across the base, mid, and apex segments. Midventricular TTS patients show significantly higher dyssynchrony in the mid segment compared to controls. Although there is an observable trend of increased dyssynchrony towards the apex, the difference is less pronounced compared to apical TTS patients, potentially due to the smaller sample size (TTS $p = 0.085$, Co $p = 0.29$).

Furthermore, the distribution of dyssynchrony in both Takotsubo groups appears to be particularly pronounced in segments 1-3 (apex to mid). This might be related to the fact that the boundaries of the affected areas ("Takotsubo area") do not precisely align with the catheter segments, leading to partial fusion of healthy and affected areas, especially in the midventricular Takotsubo group (segments 1-2; apex). This boundary issue could result in mixed signal areas that affect the dyssynchrony measurements.

To reflect these adjustments, we have added the following text to the results section on pages 4 to 5, lines 116 to 123,

"Sub-analyses of Takotsubo phenotypes (apical [$n = 14$] vs. midventricular [$n = 8$]) reveal that apical Takotsubo significantly influenced our overall findings (Supplementary Figure). These patients exhibited notable dyssynchrony in segments 1-3 (apex to mid) compared to corresponding segments in controls and significant intersegmental asynchrony between segments 1-3 and 4-5 (mid to base). In contrast, midventricular Takotsubo patients also showed significant dyssynchrony in segment 3, though the intersegmental differences were less pronounced, likely due to the smaller sample size (Supplementary Figure)."

Reviewer comment 2:

How was the Stroke Volume calculated? Using volumes or LVOT vti? I would expect lower stroke volumes in TTS patients. How the authors explain the no difference between the two groups?

Response:

Stroke volume was calculated using angiographic methods. Specifically, we used angiographic volumes to calibrate the conductance signal. This approach allows for precise measurement of stroke volume by considering changes in left ventricular volumes during the cardiac cycle. There is substantial literature supporting the use of angiography to measure ventricular volumes. For instance, Dodge and Sheehan (JACC, 1983; doi:10.1016/s0735-1097(83)80012-6) and Dodge and Baxley (Am J Cardiol, 1969; doi:10.1016/0002-9149(69)90006-x) discuss the efficacy of angiographic calibration in obtaining accurate left ventricular volume measurements.

We chose angiographic calibration given the emergency and acute settings of our patient examinations, where immediate assessment was necessary, and the technical feasibility of the procedure was paramount. To account for these methodological details, we have added the following text to the methods section on page 11, lines 333 to 338:

“Given the emergent and acute settings under which our Takotsubo patients were examined, often during nighttime conditions in the cardiac catheterization laboratory, angiographic calibration was chosen due to its rapid and reliable measurement capabilities. This method allowed simultaneous calibration with the conductance measurements, aligning with the need for immediate assessment under these conditions.”^{18,19}

Regarding the similarity in stroke volumes between Takotsubo patients and controls, this can be attributed to the compensatory mechanisms at play in Takotsubo syndrome. Despite the reduced myocardial contractility observed in Takotsubo patients, resulting from their inability to utilize the Anrep effect (as recently described in Sequeira V et al., Circ Res 2023, doi: 10.1161/CIRCRESAHA.123.323173), the heart compensates through the Frank-Starling mechanism to maintain stroke volume. This is achieved through increased left ventricular end-diastolic pressure (LVEDP) and volume (LVEDV), as noted in Stiermaier et al., J Am Coll Cardiol. 2023 (doi: 10.1016/j.jacc.2023.03.398).

In essence, Takotsubo patients manage to preserve their stroke volume through compensatory increases in LVEDP and LVEDV, despite having diminished contractile function. This compensatory response helps explain the observed similarity in stroke volumes between the two groups, despite the underlying differences in cardiac function.

We have added the following text to our results section on page 11, lines 353 to 354:

“Indicators of contractility included Ees and dP/dtmax, while effective arterial elastance (Ea) was used as an indicator of afterload.”

We have added the following text to our results section on page 4, lines 93 to 95:

“Notably, this decline in contractility occurs even under comparable afterload conditions, as reflected by unaltered effective arterial elastance (Ea) (Table 1).”

In our discussion on page 6, lines 182 to 187:

“Despite the anticipated lower stroke volumes in Takotsubo syndrome due to decreased contractility our findings showed no significant difference between the stroke volumes of Takotsubo patients and controls (Table 1). This can be explained by the compensatory utilization of the Frank-Starling mechanism in Takotsubo patients, leading to increased LV end-diastolic pressure and volume, which helps maintain stroke volume despite impaired contractility.”⁴

And on page 7, lines 203 to 207:

“In our analysis of Takotsubo patients, we observed a reduced time to Ees (Table 1) and decreases in both dE/dt_{max} (Figure 4C) and dE/dt_{min} (Figure 4D), as well as a decrease in contractility expressed by Ees with the same afterload (see Ea, Table 1) compared to controls, possibly indicating an inability to utilize the Anrep effect effectively^{7,8}.”

Additionally, we have updated **Table 1** to now include measurements of afterload (Ea, effective arterial elastance) to further substantiate this point.

REVIEWERS' COMMENTS:

Reviewer #1 (Remarks to the Author):

Many thanks for addressing each of my comments in a systematic, well-thought, thorough, and polite manner. May I point out a very slight (possibly typographic) error? In the corrections documented forwarded to me as a reviewer (2546_2_rebuttal_69571_sfmjr.pdf), rebuttal to comments 8 and 9 included the caption to Figure 3 and the following text [capitals my emphasis]:

"Although the second heartbeat for both subjects starts simultaneously, as evidenced by the dashed line marking $t=0$, the Takotsubo patient exhibits SLOWER rates of pressure change (time dp/dt_{max} to dp/dt_{min} , depicted with dashed vertical red lines)"

However the "merged" document forwarded to me ("2546_2_merged_1719306508.pdf") includes this:

"Although the second heartbeat for both subjects starts simultaneously, as evidenced by the dashed line marking $t=0$, the Takotsubo patient exhibits LOWER rates of pressure change (time dp/dt_{max} to dp/dt_{min} , depicted with dashed vertical red lines)"

Unfortunately SLOWER and LOWER have potentially opposite meanings here. Lower times are faster, slower times are higher. Additionally, regarding the 'rates of pressure change' - isn't this simply the value of dP/dt , not the time between $(dP/dT)_{max}$ and $(dP/dt)_{min}$?

For clarity, may I suggest:

"Although the second heartbeat for both subjects starts simultaneously, as evidenced by the dashed line marking $t=0$, the time from $(dp/dt)_{max}$ to $(dp/dt)_{min}$ is shorter in the Takotsubo patient".

(Thank you again for placing the vertical dashed lines - very helpful!)

Given the likely correction (if present) of this extremely minor point, I am happy to commend this article to the editors for publication, and look forward to seeing it in press soon. Congratulations to the authors!

Reviewer #2 (Remarks to the Author):

The authors have adequately addressed my previous comments that have improved the overall quality of the paper

Reviewer #1:

Many thanks for addressing each of my comments in a systematic, well-thought, thorough, and polite manner. May I point out a very slight (possibly typographic) error? In the corrections documented forwarded to me as a reviewer (2546_2_rebuttal_69571_sfmjr.pdf), rebuttal to comments 8 and 9 included the caption to Figure 3 and the following text [capitals my emphasis]:

"Although the second heartbeat for both subjects starts simultaneously, as evidenced by the dashed line marking $t=0$, the Takotsubo patient exhibits SLOWER rates of pressure change (time dp/dt_{max} to dp/dt_{min} , depicted with dashed vertical red lines)"

However the "merged" document forwarded to me ("2546_2_merged_1719306508.pdf") includes this:

"Although the second heartbeat for both subjects starts simultaneously, as evidenced by the dashed line marking $t=0$, the Takotsubo patient exhibits LOWER rates of pressure change (time dp/dt_{max} to dp/dt_{min} , depicted with dashed vertical red lines)"

Unfortunately SLOWER and LOWER have potentially opposite meanings here. Lower times are faster, slower times are higher. Additionally, regarding the 'rates of pressure change' - isn't this simply the value of dP/dt , not the time between $(dP/dT)_{max}$ and $(dP/dt)_{min}$?

For clarity, may I suggest:

"Although the second heartbeat for both subjects starts simultaneously, as evidenced by the dashed line marking $t=0$, the time from $(dp/dt)_{max}$ to $(dp/dt)_{min}$ is shorter in the Takotsubo patient".

(Thank you again for placing the vertical dashed lines - very helpful!)

Given the likely correction (if present) of this extremely minor point, I am happy to commend this article to the editors for publication, and look forward to seeing it in press soon. Congratulations to the authors!

Response:

Thank you very much for your kind words and for your valuable help in improving our manuscript.

It was indeed a typographic error. In accordance with your suggestion, we have adjusted the figure legend in lines 472 to 477 as follows (newly added text is underlined in light blue, and sections removed for clarity are struck through):

"Although the second heartbeat for both subjects starts simultaneously, as evidenced by the dashed line marking $t=0$, the ~~Takotsubo patient exhibits lower rates of pressure change (time dp/dt_{max} to dp/dt_{min} (depicted with dashed vertical red lines)~~ is shorter in the Takotsubo patient. In addition, decreased end-systolic elastance (E_{es} , depicted with dashed vertical red lines) and a shortened time to reach end-systolic elastance (dT_{es} , depicted with horizontal dotted black lines) are observed despite a lower heart rate."